# Endothelial Dysfunction: Molecular Mechanisms and Therapeutic Strategies in Kawasaki Disease

**DOI:** 10.3390/ijms252413322

**Published:** 2024-12-12

**Authors:** Lucia Paolini, Fiorentina Guida, Antonino Calvaruso, Laura Andreozzi, Luca Pierantoni, Marcello Lanari, Marianna Fabi

**Affiliations:** 1Specialty School of Paediatrics, Alma Mater Studiorum, University of Bologna, 40139 Bologna, Italy; lucia.paolini2@studio.unibo.it (L.P.); antonino.calvaruso@studio.unibo.it (A.C.); 2Pediatric Emergency Unit, IRCCS Azienda Ospedaliero-Universitaria di Bologna, 40139 Bologna, Italy; fiorentina.guida@unibo.it (F.G.); laura.andreozzi@aosp.bo.it (L.A.); luca.pierantoni@aosp.bo.it (L.P.); marcello.lanari@unibo.it (M.L.); 3Department of Medical and Surgical Sciences, Alma Mater Studiorum, University of Bologna, 40139 Bologna, Italy

**Keywords:** endothelial dysfunction, Kawasaki disease, endothelial cells, vasculitis, coronary artery aneurysms

## Abstract

The endothelium plays a key role in regulating vascular homeostasis by responding to a large spectrum of chemical and physical stimuli. Vasculitis is a group of inflammatory conditions affecting the vascular bed, and it is known that they are strongly linked to endothelial dysfunction (ED). Kawasaki disease (KD) is one childhood systemic vasculitis, and it represents the leading cause of acquired cardiac disease in children due to coronary damage and subsequent cardiovascular (CV) morbidity and mortality. We aimed to focus on the actual knowledge of ED in the pathogenesis of KD and its practical implications on therapeutical strategies to limit cardiovascular complications. Understanding ED in KD provides insight into the underlying mechanisms and identifies potential therapeutic targets to mitigate vascular damage, ultimately improving cardiovascular outcomes in both the acute and chronic stages of the disease. However, research gaps remain, particularly in translating findings from animal models into clinical applications for cardiovascular lesions and related morbidity in KD patients.

## 1. Introduction

The endothelium is a single layer of cells that lines the inner surface of blood vessels, remaining in direct contact with the blood and circulating cells. It plays a crucial role in maintaining vascular homeostasis, with the ability to respond to chemical and physical stimuli, acting as a signal transducer and producing endothelium-derived factors [1,2]. It is involved in several tasks, participating in vascular tone regulation, cellular adhesion, thrombosis, smooth muscle cells’ proliferation, and vessel wall inflammation.

There are multiple endothelium-derived factors where nitric oxide (NO) plays a pivotal role [3]. A normal vascular state can be seen as a quiescent state mainly controlled by NO’s ability to silence inflammation, cellular proliferation, and thrombosis. Laminar flow is pivotal in maintaining a quiescent, NO-controlled, endothelial state as well [4]. Moreover, physiologically, tissue-plasminogen-activator (tPA) is produced and promotes endogenous fibrinolysis [5].

From this point of view, endothelial dysfunction (ED) can be seen as a switch of state, an activation cascade towards redox signaling. It results in chemokines, cytokines, and adhesion molecules release (necessary for platelet adhesion and aggregation, leukocyte adhesion, and migration), ultimately leading to vasoconstriction, prothrombosis state, arterial stiffness, and smooth cells proliferation [6,7]. CV risk factors are all associated with ED, such as accelerated atherosclerosis, hypertension, diabetes, and ischemic heart diseases [1,8,9,10,11].

Vasculitis, a broad term that indicates a group of inflammatory conditions affecting blood vessels, has a well-established connection with ED, now considered an active player in the pathogenesis of the disease [12]. Kawasaki disease (KD) is a childhood systemic vasculitis that primarily affects small and medium-sized vessels in the prescholar aged-child, whose major complication is the involvement of coronary artery aneurysms (CAAs) in severe cases [13,14,15]. KD represents the leading cause of acquired cardiac disease in children in industrialized countries and it causes about 5% of causes of acute chest pain in adults under 40 years of age. When CAAs develop, the prognosis of the disease worsens, as coronary involvement increases the risk of both short- and long-term CV complications, including ischemic heart disease, myocardial infarction, and sudden death [16].

Shedding a light onto endothelial biology has led to the development of clinical tests that evaluate functional properties of both quiescent/normal and activated endothelial cells (ECs) [1]. Indeed, endothelial health/function can be measured via flow-mediated dilation (FMD) and nitroglycerin-mediated (NMD) dilation tests, and through the measurement of blood circulating markers [1,17,18]. Additionally, applanation tonometry enables the assessment of vascular stiffness through the analysis of pulse wave velocity (PWV) and beta stiffness index (SI), while carotid intima-media thickness (cIMT) is easily objectivated by echographic imaging [19,20,21,22].

FMD evaluates endothelial function by stimulating the release of NO from ECs, assessing its bioavailability and its ability to influence the vascular tone, thromboregulation, cell adhesion, and proliferation [1,18]. The release of NO from healthy ECs leads to vasodilation with an opposite vasoconstrictive response in ED. The NMD dilation test, instead, is evaluated through the sublingual administration of nitroglycerine and measures endothelium-independent vasodilation [17].

Lastly, applanation tonometry analysis does not directly measure endothelial cell vasoactive molecules but can document the indirect effect of arterial ED. SI study reflects the ability of beta-2-agonists to reduce arterial stiffness through NO release, indirectly measuring NO bioavailability and endothelial cell integrity. PWV measures arterial stiffness and can be acquired through different techniques: via tonometric sensors, piezoelectric mechanotransducer, ultrasound, and MRI [19,20,21,22]. Conversely, cIMT evaluates structural alterations in the vascular structure [22]. These parameters are closely associated with ED, as their elevation in childhood is linked to a higher risk of developing CV diseases in the future [23].

In this review, we intend to summarize the current understanding of molecular mechanisms underneath ED, particularly focusing on KD, the leading cause of CV morbidity and mortality in children. Moreover, we will overlook the current state of therapeutic strategies for ED, potentially contributing to restore endothelial healthy status.

## 2. Methods

For this review, Pubmed was the research engine used. A literature research for scientific articles and reviews using free text words and Advanced PubMed search was conducted, involving MeSh terms such as ED, all its abbreviation and synonyms, and KD. Endothelial dysfunction, Kawasaki disease, molecular dysfunction, and vasculitis alone and combined with “AND” were browsed.

## 3. Discussion

### 3.1. Molecular Mechanisms of Endothelial Dysfunction in KD

Recent efforts have been made to better understand the molecular mechanism underlying the pathogenesis of KD and the best alternative therapeutic options. Indeed, despite being far from being fully clarified, pathogenesis of KD is multifactorial, where genetic susceptibility is a well-established risk factor along with infectious and environmental triggers [24,25,26].

Looking deeper into the histopathological–molecular changes in KD, the activation and damage of ECs are central to its pathogenesis and are tightly linked to long-term CV complications [24].

The progression of coronary vasculopathy in KD occurs in three interconnected stages: necrotizing arteritis (NA), subacute and chronic (SA/C) vasculitis, and luminal myofibroblastic proliferation (LMP). NA is an acute, self-limiting process beginning in the endothelium and extending outward to the adventitia of medium-sized muscular and elastic arteries, characterized by neutrophilic infiltration. SA/C vasculitis involves non-neutrophilic inflammation characterized by lymphocytes, plasma cells, eosinophils, and macrophages. This stage starts in the adventitia or perivascular tissue and progresses toward the intima, closely linked with LMP. Lastly, LMP stimulates the proliferation of smooth muscle cell-derived myofibroblasts, leading to the narrowing of the vessel lumen. This process can result in vascular stenosis and is associated with severe CV outcomes, such as impaired blood flow and subsequent increased risk of ischemia, myocardial infarction, and sudden death [15,24,25,26,27].

In the acute phase of KD, perivasculitis predominantly targets the microvessels and the vasa vasorum, while the media layer of vessel is relatively unaffected. Around ten days after fever onset, the damage to the internal elastic lamina begins, leading to vascular structure loss and aneurysms formation [28]. At later stages, in response to acute inflammation, the activation of innate and cell-mediated immunity, along with increased oxidative stress, triggers the production of ECs adhesion molecules. These molecules attract leukocytes to the vascular wall [29,30,31], leading to EC edema and ED [32].

Among the inducible cell adhesion molecules involved in KD pathogenesis, intercellular adhesion molecule 1 (ICAM-1) and Vascular cell adhesion molecule 1 (VCAM-1) play an important role in mediating leukocyte migration from the vascular compartment into ECs [33]. Indeed, the upregulation of ICAM-1 and VCAM-1, mediated by inflammatory cytokines and cellular stress, has been related to the leukocyte-induced ED, vascular thrombosis, and disruption of vascular walls observed in childhood vasculitis and KD [33,34]. A peculiar genetic background to develop KD and CAA is linked to genetic polymorphisms of some alleles of HLA-B, such as HLA B51, HLA-B35, HLA-B75, HLA-Cw09 and HLA-Bw22 [35,36,37], Fc-receptors, Fc-gamma 3A (FCGR3A) [38,39,40], chemokine receptor gene-cluster (CC) haplotypes [41,42], and polymorphism of the IgG receptor IIIa [43]. These factors upregulate the expression of adhesion molecules, thus favoring inflammatory cell adhesion and local injury.

A significant contribution to ED in KD is sustained by cell-mediated immunity. An additional polymorphism predisposing to KD involves inositol 1,4,5-triphosphate 3-kinase (ITPKC) [44]. The ITPKC gene encodes a kinase implicated in the activity of nuclear factors of activated T cells (NFAT) [45,46], acting as a negative regulator of activated T cells. The polymorphisms of the ITPKC in KD are associated with increased T cell activity, leading to further damage to ECs, and contributing to the vascular complications observed in KD [13]. KD patients exhibit high levels of T helper cells (Th17) and reduced regulatory T cells (Treg), leading to hyperinflammation and immune dysregulation with subsequent increased interleukin levels [47]. Interestingly, higher serum levels of Th17 were documented in IVIG-resistant patients [48,49] at higher risk for CAA. IL-17, a proinflammatory interleukin produced by various immune cells, including Th17 cells, is overexpressed in patients with coronary artery damage, presumably because it contributes to inflammatory damage of the vascular wall [50]. Lastly, endothelial damage in KD has also been linked to increased expression of CD40L on T cells, a key marker of immune activation [51].

In addition to T-cell-mediated ED, further dysfunction is due to innate immune response, which is activated during early stages of KD. Genetic predisposition also impacts the activation of neutrophils and monocytes via the FCGR pathway, particularly FCGR2/3. Indeed, the FCGR2A-p.166His single nucleotide polymorphism is strongly associated with the development of KD [38]. The FCGR2/3 genes encode for human FCGR, crucial proteins that bridge the adaptive and innate immune systems. The upregulation of this receptor family during the acute phase of KD contributes to the increased activation of monocytes and neutrophils, thus enhancing the abnormal inflammatory response that leads to ED and arterial injury [39,40].

The activation of neutrophils results in a significant increase in serum levels of interleukin (IL)-1, IL-17, IL-6, IL-23, Chemokine (CC) ligand 3 (CCL3), and tumor necrosis factor-alpha (TNF-alpha), leading to the disruption of endothelial cell homeostasis [41,42,43,44]. CCL3L1, a potent ligand for CCR5, mediates the recruitment of leukocytes to sites of inflammation and ECs. Interestingly, patients with copy number duplications of the CCL3L1 gene show a greater susceptibility to developing KD [52].

Among cytokines, IL-1 has been seen to have a major damaging effect on coronary arteries ECs in KD [53], whereas IL-6 activates megakaryocyte maturation leading to thrombocytosis. Moreover, IL-6 may also cause ED in KD through the production of antiendothelial cell antibodies (AECAs). These antibodies play a critical role in the pathogenesis of autoimmune diseases, leading to vasculopathy and promoting proinflammatory cytokines release. Through the upregulation of adhesion molecules, AECA facilitates leukocyte adhesion to ECs, thus inducing a hypercoagulable state that further disrupts the endothelial structure [53]. Lastly, AECA promotes ECs apoptosis via an NO-mediated mechanism involving mitochondrial pathways [8]. Indeed, NO is involved in regulating both endothelial cell survival and apoptosis [8,53,54,55] (Figure 1).

Inflammasome is another player of the innate immune that promotes inflammation and induces an inflammatory form of programmed cell death, pyroptosis. It also plays a role in damaging endothelial function in KD [56]. Typically, the NLR family pyrin domain containing 3 (NLRP3) inflammasome stimulates the production of proinflammatory cytokines, reacting to abnormal molecular patterns in the human organism, such as pathogen-associated molecular pattern molecules (PAMPs) and damage-associated molecular pattern molecules (DAMPs). In KD, as well as other childhood vasculitis, an aberrant activation of NLRP3 has been linked to enhanced vascular inflammation and subsequent ED [56].

An additional mechanism affecting endothelial status in KD relates to the increased production of vasoconstrictive molecules together with a concurrent inhibition of vasodilatory activity. Elevated plasma levels of asymmetric dimethylarginine (ADMA), an endogenous inhibitor of NO synthase, have been observed in KD. The subsequent reduction in NO availability reduces vasodilation and promotes vasoconstriction, thereby exacerbating ED [57]. Furthermore, elevated plasmatic concentrations of soluble E-selectine and endothelin (ET-1), two potent vasoconstrictor molecules, have been significantly associated with the progression of childhood vasculitis [58]. Particularly, activation of the subfamily of ET(A) receptor of ECs by ET-1 leads to vascular smooth muscle cell vasoconstriction. This mechanism, along with decreased production of NO plasmatic levels induced by vascular inflammation, further promotes ED and arterial stiffness [2,59]. Consistent with these findings, KD patients, particularly those with CAA, showed increased arterial stiffness, as indicated by elevated SI and PWV [2,20,21], along with reduced flow-mediated vascular dilation, compared to healthy controls. However, nitroglycerin-mediated dilation usually remains unaffected. This suggests that vascular smooth muscle impairment may occur at a later stage of the disease [60].

He et al. identified a fundamental mechanism of ED contributing to vascular wall injury and aneurysm formation in KD [61]. The proinflammatory state of KD prompts ECs to undergo a transition to a mesenchymal phenotype via the Transforming Growth Factor beta (TGF-β) pathway [62]. This shift results in the expression of spindle-shaped cells within the vascular media, which attract proinflammatory molecules and compromise the integrity of vascular walls. At the molecular level, this transition is associated with a decreased expression of key endothelial markers, including endothelial nitric oxide synthase (eNOS) and vascular endothelial cadherin (VE-cadherin) [54]. The role of TGF-β in the development of CAA has been recently confirmed by genome-wide association studies [62]. Nucleotide polymorphisms in genes associated with the TGF-β pathway, including TGF-β2, TGF-β receptor 2 (TGFβR-2), and SMAD3, have been linked to an increased risk of developing KD and CAA, as polymorphisms related to TGF-β downregulate the Treg immune response [63]. Moreover, activated TGF-β is implicated in T cell activation and cardiovascular remodeling, both of which are active processes in the development of KD.

Lastly, microRNAs (miRNAs) have been identified as key players in the development of ED, particularly through their links to inflammation and oxidative stress. MiR-27b, a prominent member of the miRNA family in ECs, promotes angiogenesis. Under TNF-alpha-induced stress, however, MiR-27b expression in ECs decreases, leading to inflammation, reactive oxygen species accumulation, and mitochondrial alterations. Conversely, MiR-27b overexpression has been shown to counteract the effects of TNF-alpha, reducing EC damage and providing a protective benefit [64]. Interestingly, MiR-27b shows potential as a biomarker for KD and may offer a promising therapeutic target for its treatment [65].

ED can be monitored. One way to achieve that is by measuring circulating bioactive molecules of endothelial origins that can be detached in the bloodstream after an injury. When the endothelium is activated, inflammatory cytokines, NO, adhesion molecules, and other small molecules are released [1]. During the acute inflammation phase, circulating endothelial cells (CECs) and endothelial progenitor cells (EPCs) can be detached in the bloodstream [66]. These cells appear to reflect an effort to restore endothelial integrity: ECs near the injury site migrate and proliferate locally, leading to elevated levels in the bloodstream, which correlate with the severity of ED. EPCs are released from the bone marrow, where they promote endothelial repair and vasculogenesis due to their capacity to differentiate into various cell types [67,68]. In KD, EPC levels are elevated, yet their functionality is impaired, leading to slower endothelial repair [69]. CECs also are increased and their level significantly grows in the 2 weeks after standard treatment with IVIG [28].

Furthermore, endothelial microparticles (EMPs), which are released by activated ECs, are significantly elevated in KD compared to controls, serving as valuable biomarkers of endothelial damage in childhood vasculitis [70]. EMPs are extracellular vesicles that bud from damaged ECs, and their formation is enhanced by proinflammatory factors such as TNF-alpha, thrombin, inflammatory cytokines, and reactive oxygen species (ROS). Given the elevated levels of these factors in vasculitis, EMPs are increasingly recognized as reliable markers of ED [28,32,71]. Moreover, asymmetric dimethylarginine, an endogenously produced competitive antagonist of NO synthase, has been largely studied and proven to be associated with ED [72]. Other well-established molecules that can be measured in the bloodstream during vascular damage include E-selectin [73], cell adhesion molecules, P-selectin, and tPA [2].

### 3.2. Therapeutic Strategies in KD

The standard treatment for KD includes intravenous immunoglobulin (IVIG) and acetylsalicylic acid (ASA) [15]. Although the precise mechanism of IVIG is not fully understood, its pharmacodynamics involve several key effects: neutralization of pathogenic autoantibodies, suppression of TNF-alpha, and modulation or reduction in both B and T cell function [28,73,74]. However, the administration of IVIG does not seem to completely reverse the ED observed in KD patients. Indeed, after IVIG treatment, the plasmatic levels of CECs—markers of ED—do not decrease [75].

ASA modulates the inflammation and prevents thrombosis and it is usually administered with IVIG [15].

The use of corticosteroids as a first-line treatment is suggested in those patients at high risk for IVIG resistance and for coronary involvement. Molecular findings on endothelial biology have brought about a better understanding of the sequence of events and bioactive molecules causing ED, providing new possible targeted treatment to improve outcomes [75]. In the blooming era of biological drugs, targeted key-cytokines have been used in clinical trials with controversial results [74,76]. Anakinra, an interleukin-1 (IL-1) inhibitor, has demonstrated effective results in managing IVIG-resistant forms of KD, reducing the incidence of CAA in treated patients [77]. In murine models, endothelial-to-mesenchymal transition (EndMT) has been identified as a critical link between inflammatory stress and ED in aortic aneurysm disease. The loss of IL-1 signaling appears to attenuate EndMT, potentially reducing aneurysmatic lesion formation [78]. Furthermore, the inhibition of IL-1β appears to enhance autophagy in ECs, a homeostatic mechanism crucial to maintaining normal cardiovascular function and the structural integrity of cardiovascular tissues. These mechanisms may contribute to explain the effectiveness of anakinra in reducing CAA in IVIG-resistant KD patients.

Another biologic drug used is infliximab, a TNF-alpha antagonist, which has proven to be another effective treatment option for IVIG-resistant KD patients. Serum TNF-alpha levels are increased in KD, potentially activating the ECs to express ICAM-1, VCAM-1, inducible nitric oxide synthase (iNOS), and IL-1β. In ECs, resveratrol, a cardioprotector via induction of autophagy, inhibited TNF-alpha-induced ICAM-1, iNOS, and IL-1β mRNA expression in human coronary ECs [79]. Inhibiting TNF-alpha with anti-TNF-alpha antagonists may help reduce ED by blocking the associated inflammatory pathway. Moreover, an experimental study showed a reduction in inflammation and cytotoxicity mediated by TNF-alpha signaling, thereby reducing ED, via miR-27b induction. MiRNAs are, in fact, potential therapeutic agents for modulating ED with possible future applications [65,80].

In the near future, novel therapeutic targets may include VCAM and ICAM, which are associated with various vascular- and inflammation-related diseases. Indeed, modulating their activity could help influence the progression of these conditions [33,81]. The role of ET in ED was highlighted in a double-blind, placebo-controlled crossover study, where ET-1 blocking pharmacological strategies led to reduced arterial stiffness and increased tPA release [82]. Additionally, NO modulators have been demonstrated to prevent AECA-mediated ECs damage and the inflammatory state [76].

Another potential therapeutic target in KD is the Nuclear Factor of Activated T Cells (NFAT) signaling pathway [45]. In a KD vasculitis mouse model, Forkhead box protein O4 (FOXO4) acts as a transcriptional repressor, helping to maintain endothelial cell homeostasis by inhibiting NFAT signaling [83]. Specifically, FOXO4 blocks NFAT’s activity by binding to its promoter region. In FOXO4-knockout mice, the inhibition of NFAT2, the most potent member of the NFAT family, via a specific peptide was shown to reduce inflammation and decrease inflammatory infiltration [83]. The study suggested that the FOXO4/NFAT2 pathway may represent a novel therapeutic target, and that leveraging its intrinsic inhibitory mechanisms could offer new treatment options for KD. Additionally, drugs that inhibit NFAT, such as cyclosporine (CsA), may prevent the progression of inflammation in the arterial wall by blocking the infiltration of cytotoxic CD8+ T cells [83]. In experimental models, ECs exposed to sera from KD patients treated with CsA exhibited reduced proliferation, angiogenesis, NFATc1 levels, and inflammatory molecule production compared to those untreated with CsA. This suggests that CsA could have cytoprotective effects, enhancing endothelial homeostasis via the Ca^2+^/NFAT pathway [45]. However, it remains uncertain whether cyclosporine has protective effects on coronary artery involvement, particularly regarding CAAs, when calcineurin inhibitors are used in refractory KD cases.

Another mechanism contributing to KD pathogenesis is pyroptosis linked to the Sirtuin 1 (SIRT1)-NF-κB signaling pathway, a proinflammatory form of programmed cell death, which can exacerbate inflammatory disorders [84]. SIRT1, an NAD+-dependent deacetylase, plays a protective role in CV inflammation by promoting the resolution of inflammation and inhibiting pathological remodeling of the myocardial tissue. Through its regulatory function in the NF-κB pathway, SIRT1 helps reduce inflammation, providing a potential protective mechanism against myocardial damage in KD [80]. Notably, NF-kB has opposite functions, inducing proinflammatory responses [85]. In addition to SIRT1, SIRT6 has been recognized for its protective roles against inflammation, vascular aging, heart disease, and atherosclerosis. At the vascular level, SIRT6 supports endothelial cell replication and vessel formation. Its downregulation during oxidative stress accelerates cell aging, triggers inflammatory pathways, and impairs EC function, underscoring its key role in aging and inflammation [86].

In addition, in cellular and murine models of KD, the effects of Forsythoside B (FTS-B) were investigated, focusing on the same pathway [84]. FTS-B inhibited key inflammatory markers such as TNF-α, IL-6, and IκB, while modulating NF-κB activity. In KD models, FTS-B treatment demonstrated protective effects, reducing SIRT1 expression and increasing NF-κB activation. It restored SIRT1 activity and inhibited NF-κB p65, thereby decreasing inflammatory damage to human coronary artery ECs. FTS-B also lowered lactic acid dehydrogenase levels, ROS, and improved cardiac function, including better ventricular function in echocardiographic assessments [84].

Another promising treatment option is the use of statins, known for their pleiotropic effects: beyond lowering cholesterol, statins have been shown to improve surrogate markers of atherosclerosis and ED. During the process of aneurysm formation, myofibroblast-like cells—spindle-shaped cells—recruit proinflammatory cells and contribute to arterial wall damage by secreting IL-17, matrix metalloproteinases (MMPs), and connective tissue growth factor (CTGF) [87]. These cells can originate from vascular ECs, perivascular progenitor cells, vascular smooth muscle cells losing differentiation markers, and fibroblasts through EndMT [61]. At the molecular level, EndMT is manifested by the induction of mesenchymal markers as well as decreased ECs markers such as vascular endothelial cadherin and eNOS. Interestingly, EndMT can be reversed by the overexpression of Krüppel-like factor 4 (KLF4), a master regulator of ECs homeostasis and phenotype [76]. Atorvastatin improves endothelial function, reduces inflammation, and activates the KLF4–miR-483 axis [56]. This activation reduces CTGF in ECs and could potentially attenuate EndMT, providing a therapeutic strategy to preserve vascular wall integrity in KD patients [56]. Additionally, the study found that lower levels of KLF4 correlated with EndMT, highlighting statins’ role in restoring balance through KLF4 upregulation [85]. Atorvastatin has also been shown to significantly improve chronic vascular inflammation and ED, as measured by FMD of the brachial artery in children with KD who developed CAA [88]. Similarly, pravastatin has demonstrated positive effects on endothelial function and reduction in low-grade chronic inflammation in KD patients with medium-to-large coronary aneurysms after six months of treatment: FMD, NMD, and carotid artery SI improved after this treatment [89]. Furthermore, levels of high-sensitive C-reactive protein, lipid profile, and EPCs were assessed. While the serum lipid profile improved, the number of circulating EPCs did not show significant differences between baseline and after the six-month treatment in both KD patients and controls [89].

## 4. Conclusions

ED has emerged as a significant pathogenic mechanism in vasculitis and in KD. Its impact is critical, as endothelial injury can lead to severe outcomes, including myocardial infarction and sudden death in both children and adults. The better knowledge of ED has dual purpose: it enhances our understanding of the underlying pathological mechanisms and, on the other hand, it provides potential targets for treatment aiming to reduce vascular damage and preventing abnormal vascular structures in both the acute and chronic stages of the disease.

However, substantial gaps remain in our understanding of this topic, particularly regarding the concept of “healthy vascular status” and the molecular and histological mechanisms that disrupt vascular homeostasis in pathological conditions like KD. Further research is essential to explore the therapeutic potential and practical application of these findings, aiming to enhance treatment strategies for vascular complications associated with KD. Translating insights from basic science and animal models into clinical practice will be crucial.

## Figures and Tables

**Figure 1 ijms-25-13322-f001:**
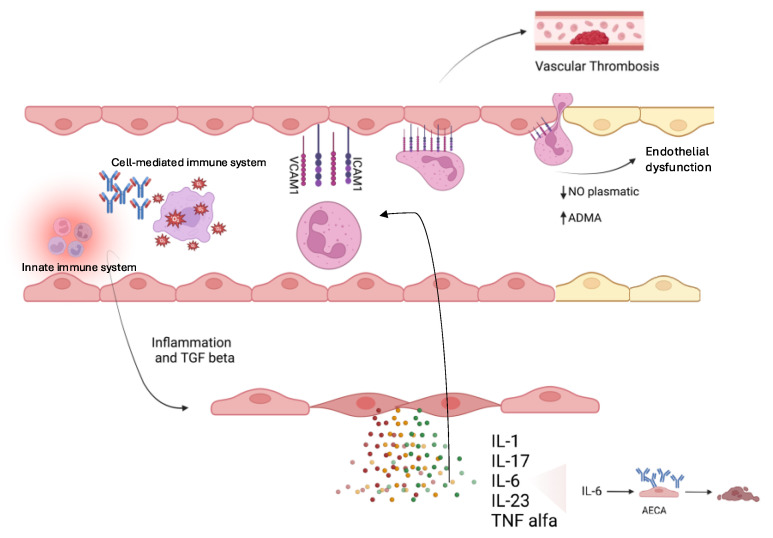
The activation of both the innate and cell-mediated immune systems leads to the production of several interleukins (e.g., IL-1, IL-17, IL-6, IL-23) and TNF-alpha. This triggers intravascular inflammation, which promotes the activation of adhesion molecules (such as VCAM), facilitating endothelial dysfunction. Subsequently, this process enhances the migration of platelets into the vascular walls, contributing to vascular thrombosis. Notably, IL-6 is associated with the generation of antiendothelial cell antibodies (AECAs), which can induce apoptosis in ECs. Additionally, a novel molecular mechanism linked to ED involves the expression of a mesenchymal phenotype through inflammatory processes and the Transforming Growth Factor beta (TGF-β) pathway. The presence of spindle-shaped cells in the vascular media further stimulates the expression of inflammatory cytokines, exacerbating endothelial damage.

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
