# Peer review of "Endothelial Dysfunction: Molecular Mechanisms and Therapeutic Strategies in Kawasaki Disease"

_ijms, 2024, doi:10.3390/ijms252413322_

Round 1

Reviewer 1 Report

Comments and Suggestions for Authors

Dear Editor,

First of all, thank you for the opportunity to write this review. The manuscript addresses an important and intensely debated topic nowadays. The idea is very good but there are some important minuses of the organization of the text:

-       It would be very useful for the authors to introduce data regarding the work methodology: the selection criteria of the articles and the result of this selection, the period during which the retrospective study was carried out. The issue related to certain predictive factors of this disease should also be addressed, such as certain genetic markers (such as HLA-B51 and HLA-Bw22j2 serotypes, chemokine receptor gene-cluster CCR2-CCR5 haplotypes and FCGR3A polymorphism of the IgG receptor IIIa). These factors play the role of upregulation of adhesion molecules.

-       It would be equally useful to introduce a prism flow diagram (Page MJ, McKenzie JE, Bossuyt PM, Boutron I, Hoffmann TC, Mulrow CD, Shamseer L, Tetzlaff JM, Akl EA, Brennan SE, Chou R, Glanville J, Grimshaw JM, Hróbjartsson A, Lalu MM, Li T, Loder EW, Mayo-Wilson E, McDonald S, McGuinness LA, Stewart LA, Thomas J, Tricco AC, Welch VA, Whiting P, Moher D. The PRISMA 2020 statement: an updated guideline for reporting systematic reviews. BMJ. 2021 Mar 29;372:n71. doi: 10.1136/bmj.n71. PMID: 33782057; PMCID: PMC8005924).

-       Since it is not an original article, I ask the authors to specify the origin of the image used or the source of inspiration.

-       The bibliography includes only 24 titles from the last 5 years out of a total of 85. Many of these references are very old and should be replaced with more current ones, if they do not represent fundamental, reference research in this field.

Author Response

Comments 1: The heading discussion should be removed after the introduction.

Response 1: Thank you, the discussion has been removed from the abstract as suggested

Comments 2: Inside the molecular mechanism of endothelial dysfunction in KD, suddenly talking about the neutrophil activation doesn't correlate with the EC dysfunctions, which looks irrelevant here, please connect it with a proper description.

Response 2: Thank you for this comment. We have revised the discussion and included additional information on endothelial dysfunction to create a clearer connection. The revised version now addresses the activation of the innate immune response first, followed by cell-mediated immunity, in the attempt to provide a clear explanation of the connection with ED.

Comments 3: Similarly, under the same section, from line 164 onwards, the author started to talk about the activation of cell-mediated immunity. It is not clear how EC dysfunction activates the immune system in KD. The authors need to correlate it properly with the explanations.

Response 3: As suggested, we’ve modified the text.

Comments 4: In lines 203 to 205, the author has mentioned the circulating EC, which is creating confusion, please clarify the sentence. 

Response 4: As suggested, we’ve modified the text.

Comments 5: There are various articles published that explain the molecular mechanism of EC in KD, thus, I do not see any novelty in this paper, except the author has added a therapeutic strategy that is not well explained and outlined. 

Response 5: We acknowledge that the literature on endothelial dysfunction in KD patients is extensive; however, our aim was not to propose novel mechanisms, but to systematically present the articles about the topic in the attempt to provide a comprehensive overview of the existing findings on the topic. As suggested, regarding the therapeutic strategy, we have rephrased the text trying to be clearer for the reader and to modify it to address the reviewer's concerns.

Comments 6: Figure provided by the author for the molecular mechanism of EC in KD is not informative.

Response 6: We have update the figure to address your consideration.

Reviewer 2 Report

Comments and Suggestions for Authors

The review article "Endothelial Dysfunction: Molecular Mechanism and Therapeutic Strategies in Kawasaki Disease" by Paolini et al. is interesting. However, it lacks cohesiveness and uses information that is not relevant to the title and heading provided in the review. Below are the few comments I observed following the review.

1. The heading discussion should be removed after the introduction. 

2. Inside the molecular mechanism of endothelial dysfunction in KD, suddenly talking about the neutrophil activation doesn't correlate with the EC dysfunctions, which looks irrelevant here, please connect it with a proper description.

3.  Similarly, under the same section, from line 164 onwards, the author started to talk about the activation of cell-mediated immunity. It is not clear how EC dysfunction activates the immune system in KD. The authors need to correlate it properly with the explanations.

4. In lines 203 to 205, the author has mentioned the circulating EC, which is creating confusion, please clarify the sentence. 

5. There are various articles published that explain the molecular mechanism of EC in KD, thus, I do not see any novelty in this paper, except the author has added a therapeutic strategy that is not well explained and outlined. 

6. Figure provided by the author for the molecular mechanism of EC in KD is not informative.

Author Response

(The authors gave the same response as above.)

Round 2

Reviewer 1 Report

Comments and Suggestions for Authors

I consider that the changes made to the manuscript by the authors qualify it for acceptance for publication.

Reviewer 2 Report

Comments and Suggestions for Authors

A revised version of the review article looks good.